# Multiplex Target-Redundant RT-LAMP for Robust Detection of SARS-CoV-2 Using Fluorescent Universal Displacement Probes

Enos C. Kline,[a] Nuttada Panpradist,[a,b] Ian T. Hull,[a] Qin Wang,[a] Amy K. Oreskovic,[a] Peter D. Han,[c,d] Lea M. Starita,[c,d] Barry R. Lutz[a,d]

[a]Department of Bioengineering, University of Washington, Seattle, Washington, USA

[b]Global Health for Women Adolescents and Children, School of Public Health, University of Washington, Seattle, Washington, USA

[c]Department of Genome Sciences, University of Washington, Seattle, Washington, USA

[d]Brotman Baty Institute for Precision Medicine, Seattle, Washington, USA

**ABSTRACT** The increasing prevalence of variant lineages during the COVID-19 pandemic has the potential to disrupt molecular diagnostics due to mismatches between primers and variant templates. Point-of-care molecular diagnostics, which often lack the complete functionality of their high-throughput laboratory counterparts, are particularly susceptible to this type of disruption, which can result in false-negative results. To address this challenge, we have developed a robust Loop Mediated Isothermal Amplification assay with single tube multiplexed multitarget redundancy and an internal amplification control. A convenient and cost-effective target-specific fluorescence detection system allows amplifications to be grouped by signal using adaptable probes for pooled reporting of SARS-CoV-2 target amplifications or differentiation of the Internal Amplification Control. Over the course of the pandemic, primer coverage of viral lineages by the three redundant sub-assays has varied from assay to assay as they have diverged from the Wuhan-Hu-1 isolate sequence, but aggregate coverage has remained high for all variant sequences analyzed, with a minimum of 97.4% (Variant of Interest: Eta). In three instances (Delta, Gamma, Eta), a high-frequency mismatch with one of the three sub-assays was observed, but overall coverage remained high due to multitarget redundancy. When challenged with extracted human samples the multiplex assay showed 87% or better sensitivity (of 30 positive samples), with 100% sensitivity for samples containing greater than 30 copies of viral RNA per reaction (of 21 positive samples), and 100% specificity (of 60 negative samples). These results are further evidence that conventional laboratory methodologies can be leveraged at the point of care for robust performance and diagnostic stability over time.

**IMPORTANCE** The COVID-19 pandemic has had tremendous impact, and the ability to perform molecular diagnostics in resource limited settings has emerged as a key resource for mitigating spread of the disease. One challenge in COVID-19 diagnosis, as well as other viruses, is ongoing mutation that can allow viruses to evade detection by diagnostic tests. We developed a test that detects multiple parts of the virus genome in a single test to reduce the chance of missing a virus due to mutation, and it is designed to be simpler and faster than typical laboratory tests while maintaining high sensitivity. This capability is enabled by a novel fluorescent probe technology that works with a simple constant temperature reaction condition.

**KEYWORDS** coronavirus, diagnostics, molecular methods

Address correspondence to Barry R. Lutz, blutz@uw.edu.

The authors declare a conflict of interest. Patent applications have been filed on several components of this assay. E.C.K., N.P., Q.W., I.T.H., D.L., and B.R.L. are inventors on one or more provisional patent applications. E.K., N.P., Q.W., I.H., A.K.O., and B.R.L. have equity in a startup company that licenses this technology.

The COVID-19 pandemic is an unprecedented crisis in the modern era, spreading across the planet in a matter of months, infecting and killing millions, while disrupting the lives of billions (1). An essential element of the response strategies to COVID-19 is diagnostic testing, which informs clinical intervention, quarantine, and epidemiological monitoring

(2). Nucleic acid amplification tests (NAATs) remain the most accurate approach for diagnosis of infectious diseases, including SARS-CoV-2 infection. However, RNA viruses like SARS-CoV-2 have a high mutational rate, which can result in elevated levels of sequence diversity accumulating as they propagate. This is a critical obstacle for NAATS because mismatches between the primer oligonucleotides and the template sequences can impair an assay and produce false-negative results. As transmission has progressed, SARS-CoV-2 has diversified in distinct lineages, each with signature mutations throughout the genome (3). The emergence of this genetic diversity has rendered some NAATs susceptible to false-negative results as a consequence of mismatches between their primers and mutations in the targeted nucleic acid, causing these tests to be altered or withdrawn by the U.S. FDA (4). This challenge posed by mutation for NAATs is not limited to SARS-CoV-2; similar phenomena have been observed for other human pathogens (5, 6).

Laboratory testing strategies to lessen this risk include redundant testing with alternative methods, diagnostic panels with multiple target regions (7), and/or primer sets with degenerate bases to account for known genetic variability (8). While degenerate primers are accessible and inexpensive, they are often limited by assay design constraints and do not account for unknown or novel mutations. Repeat and multiple testing is an effective strategy, but requires additional resources, labor, and complexity of design or implementation. These considerations are manageable in contemporary diagnostic laboratories but can be prohibitive in lower resource settings. Nearly all laboratory assays for SARS-CoV-2 use redundant targets to mitigate mutations and an internal control to account for sample processing or interference.

A critical aspect of the Centers for Disease Control and Prevention's (CDC) Strategy for Global Response to COVID-19 (2020 to 2023) is augmenting our current ability to rapidly identify COVID-19 infections so that the chain of transmission can be disrupted. Essential to this effort is the development of diagnostics that can be performed at the point of care (POC) that minimize the time to result (TTR) of the test and are deployable in otherwise underserved populations. These settings are inherently "low resource," and necessitate diagnostic methods with simplified chemistry, hardware, and limited sample processing relative to the standard of practice for molecular diagnostics, polymerase chain reaction (PCR). Advancements in isothermal nucleic acid amplification technologies over the past 3 decades largely satisfy these constraints while still providing high sensitivity. This has led to a boom in isothermal amplification technologies and NAATs based on them (9). Despite their advantages, there are some areas where the isothermal NAATs are lacking compared to PCR. Single-pot multiplexing has been infrequently demonstrated despite being a prerequisite for internal amplification control (IAC) systems and useful for multiple target redundancy (10–12) In this work we look to contribute to this capability as it relates to Loop Mediated Isothermal Amplification (LAMP) for the detection of SARS-COV-2.

Herein we describe a multiplex reverse transcriptase LAMP (mRT-LAMP) combining three assays, each targeting a unique region of the nucleocapsid (NC) gene, and an IAC assay to validate diagnostic viability with a negative result. To accomplish this a target-specific detection mechanism is required to differentiate target and IAC amplifications. A variety of fluorescence probe systems have been previously described (13) and remain an ongoing area of LAMP innovation. We have designed a universal target-specific fluorescence probe system that is flexible and easy to implement in an existing LAMP. In this method, engineered adapter sequences are incorporated into the LAMP amplicons which then serve as a template for detection by displacement probes. The resulting assay chemistry is sensitive, specific, and durable while simplifying the development process. We evaluate the limit of detection, cross-reactivity with other organisms, and reactivity with extracted RNA from patients infected with SARS-CoV-2. This effort serves as the molecular assay basis for our development of a POC diagnostic platform for SARS-CoV-2 (14).

## RESULTS

To efficiently combine mRT-LAMP assays and differentiate between target and IAC amplification in a crude sample matrix requires two key features: a target-specific

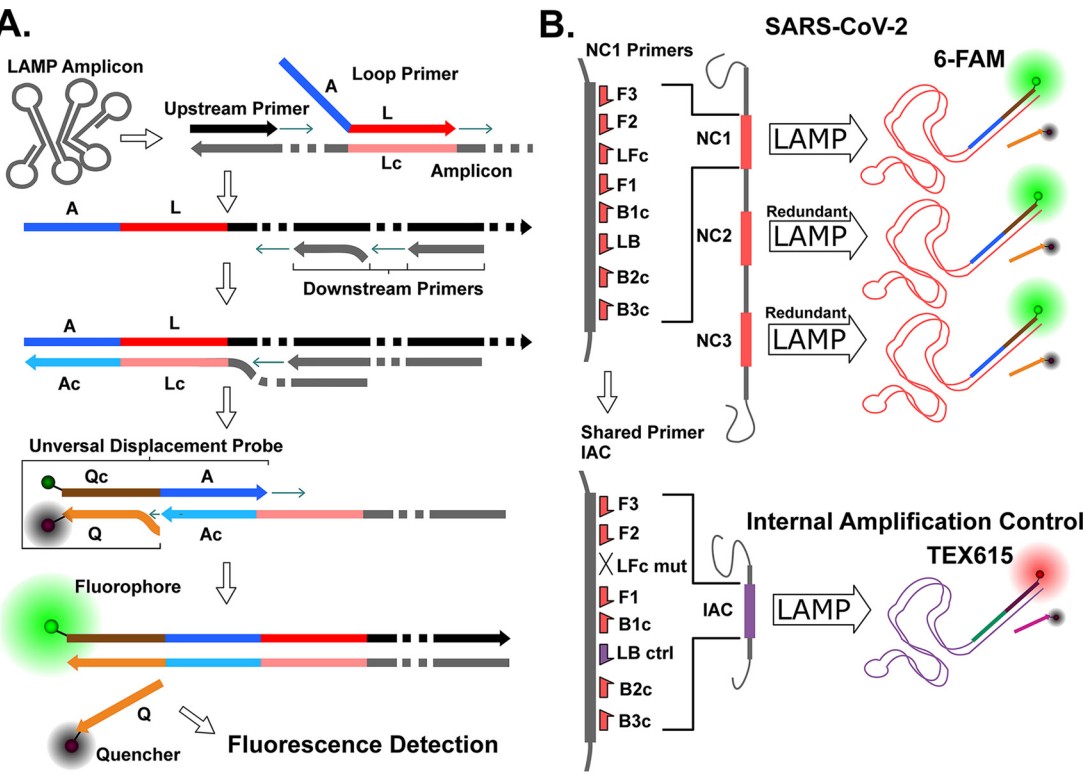

**FIG 1** Multiplex RT-LAMP (mRT-LAMP) fluorescence detection by Universal Displacement Probes (UDP). (A) UDP incorporation during LAMP amplification and activation by displacement of quenching strand. Primer and probe refer to loop (L), adapter (A), and quencher (Q), with complementary sequences denoted with the suffix "c" (e.g., "Lc" is the reverse complement "L"). (B) Two-channel fluorescence detection of multiplexed redundant LAMP products (6-FAM) and shared-primer IAC (TEX615) by UDPs. Primer designations refer to forward (F), backward (B), and loop (L) using conventional LAMP terminology.

probe technology (Fig. 1) and a strand displacement polymerase with low nontemplate amplification. We developed fluorescent universal displacement probes (UDPs) to allow multiplexed assays to be combined or parsed into fluorescence channels with a minimum number of probes. UDPs themselves are engineered sequences that use a universal adapter sequence on a loop primer for target-specific detection (Fig. 1A). In the configuration presented here, three independent SARS-CoV-2 targets are designed to report to a single green (6-FAM) fluorescent probe, and the IAC is designed to report to a red (TEX615) fluorescence channel (Fig. 1B). We previously developed an in-house thermostable strand displacement polymerase (TFpol) with very low nonspecific amplification that is amenable to multiplexing. The TFpol design was inspired by the chimeric polymerase method of Morant (15) using the polI polymerase of *Thermus Thermophilus* as the backbone, an enzyme shown to be tolerant of many polymerase inhibitors (16, 17). UDPs and TFpol combine to allow for a flexible and robust mLAMP system, compatible with multiple target redundancy, IAC controls, and potential for reduced sample preparation.

**Analytical performance of SARS-CoV-2 mRT-LAMP.** Functionality of the individual redundant targets in the mRT-LAMP was verified using synthetic RNA fragments corresponding to NC1, NC2, or NC3 mRT-LAMP assay footprints. All three target regions generated detectable amplification (Fig. S1A) with similar average reaction times with 200 copies of transcript RNA (NC1: 26.4 min, NC2: 26.3, NC3: 28.7 min; Fig. S1B).

The multiplex assay was evaluated with synthetic target RNA containing all three target regions in the presence of $10^5$ copies of a single-stranded DNA internal amplification control (Fig. 2A). The amount of IAC was chosen to allow detection of low-copy targets prior to detection of the IAC, in order to reduce resource competition between

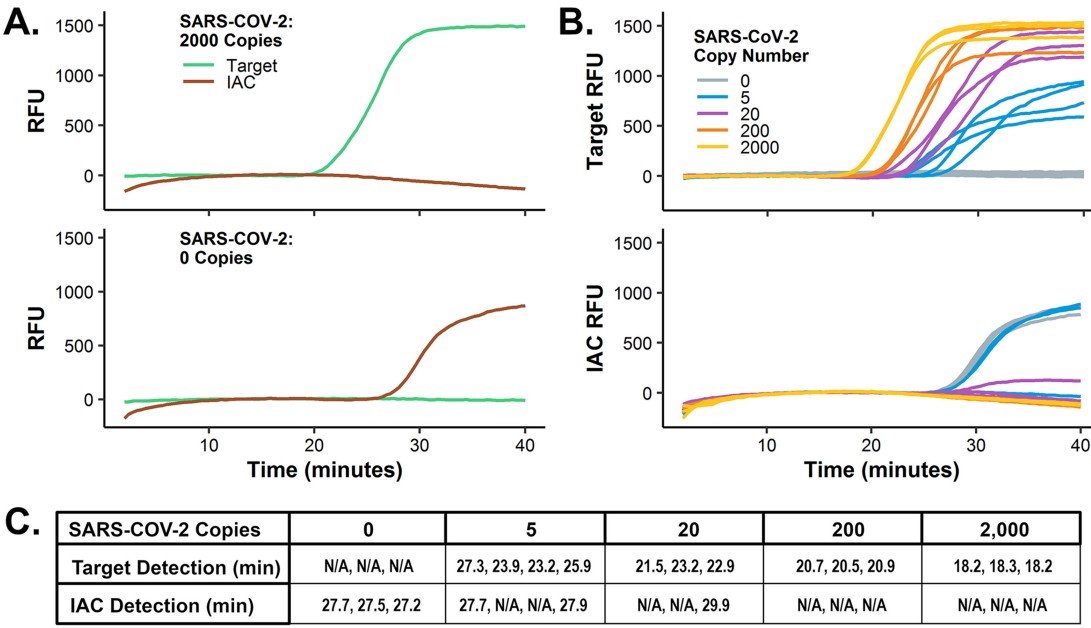

**FIG 2** Analytical performance of mRT-LAMP for SARS-CoV-2. (A) Characteristic amplification of multiplexed SARS CoV-2 targets and internal amplification control (IAC) with real-time fluorescence detection by universal displacement probes (UDP). Single representative run with 200 copies of synthetic RNA input or a no template control (NTC). (B) Analytical sensitivity of multiplexed SARS-CoV-2 target and IAC. IAC amplifications (bottom) correspond to target amplifications (top). Target synthetic RNA input: 2,000 ($n = 3$), 200 ($n = 3$), 20 ($n = 3$), 10 ($n = 3$), or 5 copies per reaction ($n = 4$); and NTC ($n = 3$). (C) Time to detect signals from SARS-CoV-2 and IAC for reactions from panel B.

target and control amplifications. This timing differential is possible because of the reduced rate of amplification with a single loop primer in the IAC primer set, compared to the target assays with a standard complement of LAMP primers. Input of 200 SARS-CoV-2 RNA copies (Fig. 2A, top) resulted in detection of green fluorescence in about 21 min, while the IAC was not detected. For zero SARS-CoV-2 input copies, there was no target amplification, and the IAC signal was detected by red fluorescence at about 27.5 min (Fig. 2A, bottom). This behavior is ideal for a shared-primer IAC strategy, permitting detection of the target organism or, alternatively, validating the assay chemistry with the control reaction in the absence of target NAs. The analytical sensitivity was assessed with synthetic RNA target (Fig. 2B). All reaction mixtures containing target RNA were positive, and all NTC reactions detected IAC amplification and were negative for target (Fig. 2A). Some IAC amplifications were detected in low copy reaction mixtures containing target RNA (Fig. 2A), and fluorescence drift of the IAC was observed, but their presence did not compromise target detection. The assay detected down to 5 copies per reaction ($n = 4$), and all reactions had threshold times of 30 min or less for both the target and IAC (Fig. 2C).

**Tolerance to transport media.** To evaluate the tolerance of the assay to potential media contaminants, a selection of commercially available co buffered transport reagents were spiked into reactions with a 25% final concentration. For a 20 $\mu$L total reaction volume, 5 $\mu$L of 1$\times$ DMEM (11965-06, Gibco), 1$\times$ VTM (BD 220527, Copan), 1$\times$ PBS (SH30256.01, GE) or 0.9% sodium chloride (diluted from 5 M stock 71386-1L, Sigma) was added into the mRT-LAMP reactions with final synthetic SARS-CoV-2 RNA of 0, 20, or 200 copies (Fig. S2). Successful SARS-CoV-2 amplification was observed for all samples containing template under all buffer conditions.

**Performance with extracted clinical specimens.** The SARS-CoV-2 mRT-LAMP was evaluated against a collection of pre-extracted patient specimens. Of the 102 samples evaluated by RT-PCR, 93 were determined to contain human origin material by positive RNase P (RP) results; all samples that were negative for RP were also negative for SARS-CoV-2 and were considered indeterminate. Of the 93 specimens verified to contain

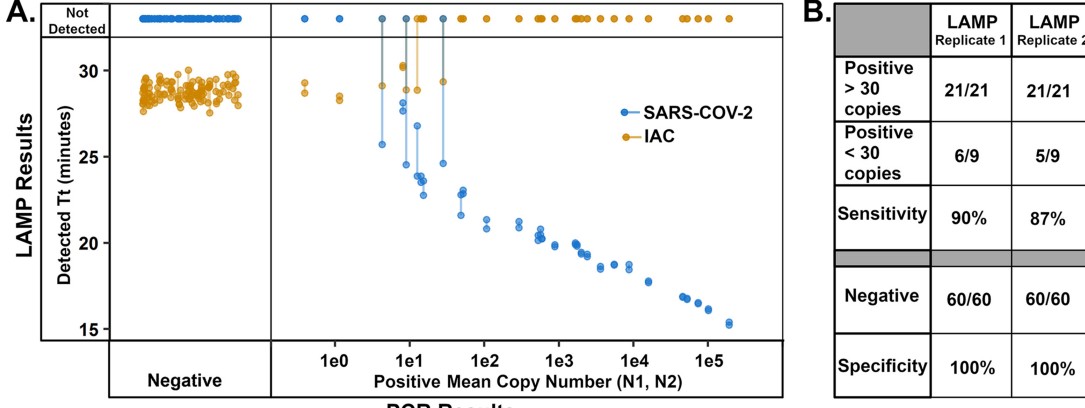

**FIG 3** mRT-LAMP amplification of extracted nasal specimens. Samples confirmed as Negative or Positive for SARS-CoV-2 by RT-PCR panel (N1, N2, RP) were amplified by duplicate mRT-LAMP reactions. mRT-LAMP signals for SARS-CoV-2 are shown in blue, and IAC signals are shown in orange with detected Threshold time (Tt) or "Not Detected" reported for both in all reactions; replicate pairs for each signal are connected by a line segment. Mean copy number was derived from qPCR results of N1, N2 PCR (see Table S3).

human material 60 were found to be negative for SARS-CoV-2 and 30 were found to be positive by both reference RT-PCR assays; these were considered "Negative" and "Positive" samples, respectively. The three remaining samples were positive for SARS-CoV-2 by one reference RT-PCR assay and negative by the second, resulting in an inconclusive classification. All samples that were indeterminate or inconclusive by RT-PCR were excluded from analysis. Negative and Positive clinical samples were run in duplicate mRT-LAMP reactions.

The mRT-LAMP was able to detect negatives with 100% specificity in both sets of replicates, with detection of the IAC but no target signal (Fig. 3). Conversely, sensitivity for the two replicates was 90% (27/30) and 87% (26/30), respectively. For samples found to have more than 30 copies/mRT-LAMP reaction by reference RT-PCR, sensitivity was improved to 100% (21/21) for both replicates. The OpenArray characterization of the verified samples found 57 of 90 validated samples contained one or more other respiratory infections; 8 of 9 SARS-CoV-2 positive samples with coinfections were correctly called as SARS-CoV-2 positive, 48 of 48 SARS-CoV-2 negative samples that were positive for other pathogens were correctly called as SARS-CoV-2 negative.

**Primer coverage analysis.** Individual primer sets (Table 1) had variability in the frequency of perfect primer matches across the VOC/VOI sequence libraries (Table 2). Presence of mismatches does not necessarily preclude assay functionality and is therefore an underestimate of realized assay coverage. Combined primer coverage, where one or more primer sets had a perfect match for the target, was high across all variants, with a minimum 97.4% (for Eta).

**Performance against variant sequences with known mismatch mutations.** To practically assess the viability of multiplex target redundancy as a strategy to mitigate diagnostic failure due to primer-template mismatches as a result of viral mutation, the sub-assays and mRT-LAMP were challenged with templates known to have mutations in target regions. Synthetic RNA templates representative of SARS-CoV-2 variants Delta (Twist Bioscience, South San Francisco, California, 104539) and Omicron (Twist Bioscience, 105204), as well as the reference genome (Twist Bioscience, 102024) (RefSeq) were selected. Delta and Omicron were chosen because of their relative importance to public health (18) and the presence of fixed mismatched mutations in the NC3 and NC1 sub-assays (Table 2), respectively. Sequence alignments with Delta sequences identified a high-frequency single-nucleotide polymorphism (SNP) mutation in that lineage (G29402T) within the NC3 B3 primer binding site. Alignments with Omicron sequences revealed two mismatches in the NC1 assay: a SNP (C28311T) in the F2 primer binding site and a 9-base

**TABLE 1** Primer, probe, and control sequences for the SARS-CoV-2 mRT-LAMP[a]

| Primer set | Sequence |
|---|---|
| **SARS-CoV-2 NC1 primers** | |
| NC1 FIP | CCACTGCGTTCTCCATTC*TTTT*CCCCGCATTACGTTTGGT |
| NC1 BIP | GCGATCAAAACAACGTCGG*TTAT*TGCCATGTTGAGTGAGAGCG |
| NC1 LF | TGGTTACTGCCAGTTGAATCT |
| NC1 LB + Target adapter | **ACCAACACCTCACATCACACATAATA**GGTTTACCCAATAATACTGCGTCTTG |
| NC1 F3 | TGGACCCCAAAATCAGCG |
| NC1 B3 | ATCTGGACTGCTATTGGTGTTA |
| **SARS-CoV-2 NC2 primers** | |
| NC2 FIP | CAGCTTCTGGCCCAGTTCCTGTGGTGGTGACGGTAAAATG |
| NC2 BIP | CTTCCCTATGGTGCTAACAAAG*T*CCAATGTGATCTTTTGGTGTATTCA |
| NC2 LF | GTAGTAGAAATACCATCTTGGACT |
| NC2 LB + Target adapter | **ACCAACACCTCACATCACACATAATA**ATATGGGTTGCAACTGAGGGAG |
| NC2 F3 | CTACTACCGAAGAGCTACCAG |
| NC2 B3 | GCAGCATTGTTAGCAGGATTG |
| **SARS-CoV-2 NC3 primers** | |
| NC3 FIP | TGTGTAGGTCAACCACGTTC*T*GCTTCAGCGTTCTTCGGA |
| NC3 BIP | GTGCCATCAAATTGGATGACAAAG*G*TTTTGTATGCGTCAATATGCTTATTCAG |
| NC3 LF + Target adapter | **ACCAACACCTCACATCACACATAATA**TCCATGCCAATGCGCGACA |
| NC3 LB | CCAAATTTCAAAGATCAAGTCAT |
| NC3 F3 | GACCAGGAACTAATCAGACAAG |
| NC3 B3 | GCTTGAGTTTCATCAGCCTTC |
| **IAC (NC1) primer** | |
| IAC FL + Control adapter | **ACCACACCTACCACCACTAATAACTAA**CTCCAGCCATCCTCACCATC |
| **SARS-CoV-2 UDP** | |
| Target (CoV) UDP Probe | FITC-CCATCAGCACCAAGACTACCCACCTCGCCACCAA**ACCAACACCTCACATCACACATAATA** |
| Target (CoV) UDP Quencher | TTGGTGGCGAGGTGGGTAGTCTTGGTGCTGATGG-Iowa Black FQ |
| **IAC UDP** | |
| Control (IAC) UDP Probe | Tex615-CCTGACCACTTCCGAACCCAACCACCTACGACAG**ACCACACCTACCACCACTAATAACTAA** |
| Control (IAC) UDP Quencher | CTGTCGTAGGTGGTTGGGTTCGGAAGTGGTCAGG – BHQ-2 |
| **IAC template** | |
| IAC ssDNA | AAT GGA CCC CAA AAT CAG CGA AAT GCA CCC CGC ATT ACG TTT GGT GGA CCC TCT GGA GTC AAT GGG TGG TGC CAG AAT GGA GAA CGC AGT GGG GCG CGA TCA AAA CAA CGT CGG CCC CAA GTT GAT CTC CAG CCA TCC TCA CCA TCG TTC ACC GCT CTC ACT CAA CAT GGC AAG AAT TAA CAC CAA TAG CAG TCC AGA TG |

[a]For primers and probes, F2/B2 sequences are underlined, nontemplate linker sequences are italicized, and adapter sequences are shown in bold.

deletion (GAGAACGCA28362) in the NC1 primer binding site. Excepting those conflicts, all other assays were perfect identity matches across their primer binding regions. RT-LAMP sub-assays were evaluated individually against 200 copies of each template (see Supplementary Information). The NC1 sub-assay failed to detect the Omicron template, while all other sub-assays and multiplex assays were successful at detection of all three templates, including Omicron (Table 3).

## DISCUSSION

This initial validation of a multiplex reverse transcription LAMP assay is a further step toward more resilient point-of-care NAAT technologies with convenient implementation and development. The assay supports robust but basic functionality with competitive sensitivity, speed, and a low complexity fluorescence detection system. Because VTM has been shown to inhibit conventional PCR strategies (19), we designed our assay to use our in-house chimeric polymerase, TFpol, which has been proven to be effective with complex samples containing various transport media. TFpol supports multiplex LAMP amplifications which have been infrequently demonstrated. These capabilities, taken together, enable features that are contemporary in high–throughput laboratory testing but more challenging in point-of care diagnostics.

Multiplexed LAMP reactions with the ability to differentiate individual products by target-specific probes enable two key aspects of robust NAAT testing: internal amplification controls and multiple target redundancy. IACs are widely accepted as a means of ensuring the sample could detect a positive result if the target result is negative, by

**TABLE 2** Coverage of variant sequences by individual and multiplex targets

| SARS-CoV-2 VOC/VOI | Primer set (% of perfect primer set alignment to 1000 sequences) | | | |
| | NC1 primers | NC2 primers | NC3 primers | Multiplex coverage (1+ primer set match) |
|---|---|---|---|---|
| Alpha | 71.8 | 90.0 | 79.9 | 100.0 |
| Beta | 96.4 | 90.4 | 95.9 | 100.0 |
| Delta | 96.7 | 96.5 | 00.0 | 99.8 |
| Epsilon | 94.6 | 93.7 | 81.2 | 99.6 |
| Gamma | 00.5 | 80.7 | 88.6 | 98.8 |
| Eta | 93.9 | 00.4 | 83.7 | 97.4 |
| Iota | 88.0 | 81.5 | 92.5 | 99.9 |
| Omicron | 00.0 | 97.6 | 98.1 | 99.9 |

verifying that the reaction chemistry was viable and not inhibited by sample contaminants or otherwise compromised (20). In the context of LAMP amplification, internal controls can impair successful target detection; the resource demands of a successful LAMP mean co-amplification of multiple products with various inputs often lead to the competitive inhibition of slower assays or of amplification at lower concentration of target (Fig. 2B and C). Presumably, this can be attributed to resource depletion of limiting reagents in the reaction mix. To address this resource competition, we devised a shared-primer internal control strategy where the performance of the IAC has been intentionally impaired by using a reduced primer set. The delayed time-to-detection of the IAC can then be further controlled by adjusting the concentration of control template, ensuring reduced competition with the target amplification.

The UDP probe system that enables differentiable detection of Target and IAC amplifications has many commonalities with existing probe systems, but has a unique feature set useful for rapid development, pooled target amplification reporting, and flexible application. Many of the previously described probe systems, such as DARQ (10) and Molecular Beacons (21) are specific to endogenous target sequence and require dedicated probes. This is also true of assimilating probes (22), the technology that is conceptually most similar to UDPs. UDPs leverage a similar design, exploiting the existing compatible functionality of the loop primers, but use an adapter intermediate so that probe sequences are not directly tied to the target sequence and can be entirely engineered. This provides several advantages: the probes can be designed to be minimally interactive with other elements of the amplification mix, they can be repurposed or adapted to new or revised designs without the need to develop a new probe, and in a multiplexed reaction multiple targets can be efficiently associated with a single reporter probe. Mediator displacement probes (23) share these properties, but do not incorporate the probe label into the amplicon which may limit some applications, requires an additional Mediator oligonucleotide, and has a more complex dually labeled stem-loop probe structure. While each of the previously described probe systems has been shown to be effective in various contexts, the suitability of UDPs to a multiplex target redundant system for a rapidly mutating and variable target with an IAC is apparent.

**TABLE 3** Detection by multiplex and sub-assays of 200 copies of representative synthetic RNA

| Template | Assay (detection events) | | | | Multiplex + IAC | |
| | NC1 | NC2 | NC3 | Multiplex | Target | IAC |
|---|---|---|---|---|---|---|
| RefSeq | 3/3 | 3/3 | 3/3 | 3/3 | 3/3 | 0/3 |
| Delta | 3/3 | 3/3 | 3/3 | 3/3 | 3/3 | 0/3 |
| Omicron | 0/3 | 3/3 | 3/3 | 3/3 | 3/3 | 3/3 |
| NTC | 0/3 | 0/3 | 0/3 | 0/3 | 0/3 | 3/3 |

Multiplex target redundancy is a defining feature of this assay design even with the grouped reporter signal. Pathogen genetic variability is an important failure mode for nucleic acid amplification tests; a SNP can result in failure or underperformance of a LAMP (24) or PCR (25), and deletions or insertions can disrupt primer binding entirely. As the COVID-19 pandemic progresses, the virus will continue to accumulate mutations and diversify, posing a challenge to NAATs used for diagnosis. An alignment of publicly available SARS-CoV-2 genomes at the time of writing reveals multiple genomes with known mutations in the primer footprints of the CDC PCR designs and a range of other published assays, suggesting that mutations are an existential problem (data not shown). The likelihood of these mutations rendering all three target amplifications ineffective simultaneously is lower than for a single assay. This principle is often incorporated in commercially available conventional laboratory based NAATs, so this capability represents a convergence of state-of-the-art diagnostic methods and POC diagnostic capabilities. Our own analysis (Fig. 2) found that, since the design of the multiplex assay in early 2020 against the NCBI reference sequence, the emergence of the many variant lineages had resulted in some high-frequency mismatches in primer binding regions of our targets. While many single nucleotide polymorphisms are likely to be tolerated by a LAMP reaction, some mutations, such as those located at or near the critical termini of primers, may interfere with diagnostic performance (26). Variants Delta, Gamma, Eta and Omicron each had fixed, or almost fixed, allele mutations in primer sets NC3, NC1, NC2, and NC1, respectively. When considered in combination as a multiplex assay, the primers still showed good overall coverage despite one of the three assays being potentially compromised. This was further reinforced by directly challenging each of the sub-assays and the multiplex with synthetic RNA templates representative of the reference, Delta, and Omicron lineages. The Delta template was detected by all sub-assays and multiplex assays despite a mismatch in the NC3 B3 primer binding region. This mismatch affected a "bumper" primer less critical to the amplification process, and occurred internally, so tolerance to this low-risk mutation was not unexpected. In contrast, a mutation of the Omicron variant is very high risk; the 9-base deletion to the critical F1 primer binding site would be expected to completely disrupt the loop-forming process essential to LAMP. As expected, the NC1 sub-assay failed to detect the Omicron template when used alone. However, the multiplex assay was unaffected in its ability to detect Omicron. This is a clear demonstration of the value that multi-target redundancy holds for viral diagnostics in the context of an actively circulating variant that rapidly emerged and became prevalent. While altering or updating a single target assay in a relevant timescale to address Omicron would be logistically challenging, and mismatch-tolerant LAMP methods (26) would still be vulnerable to such a large deletion, a multitarget assay is robust to the threat. In this scenario, the affected primers could be redesigned to be inclusive to the variant, potentially, including other mismatch tolerance strategies, and phased into the multiplex assay without disruption to the diagnostic and ensuring no lapse in coverage over time.

In order to fully realize a field ready POC assay, additional development is planned. The preliminary testing performed suggests that the system is tolerant to inhibitors that might typically interfere with a direct-to-amplification workflow. However, the clinical specimens evaluated here were processed by RNA extraction. In-amplification sample lysis and testing with human sample matrix is necessary to further validate this strategy. Assessment of direct detection of contrived samples in nasal matrix and appropriate crude clinical samples is a crucial aspect of ongoing work and was explored in our companion publication (14) focused on a POC detection platform. This is particularly important for understanding the role of RNAses on assay sensitivity. With the current ssDNA IAC design sampling efficacy and RNA integrity are not assessed and are essential functions for a complete diagnostic control system. Future iterations will address this by implementing an endogenous human control target and/or encapsidated RNA, such as MS2 coliphage for this purpose. While this proposed design and chemistry is amenable to adaptation to most contemporary high-throughput NAAT

platforms as is, it is probably best leveraged in a mobile low-resource platform. Sampling, storage, and portable device solutions with additional clinical and experimental evaluation are already under development (14). These advancements will allow us to eschew infrastructure requirements that have acted as a bottleneck in current testing efforts, and when combined with the robust multiplex chemistry presented here, could act as a practical solution for decentralized testing.

## MATERIALS AND METHODS

**Preparation of TFpol polymerase.** Plasmid preparation and protein expression and purification were performed as previously described (14).

**Primer and IAC design.** Three sets of LAMP primers (Table 1) targeting different regions of the SARS-CoV-2 nucleocapsid phosphoprotein were designed manually using the primer design feature of Geneious 8.1.9 (27) against the SARS-CoV-2 reference sequence (GenBank accession number: NC_045512). IDT OligoAnalyzer (28) and NUPACK (29) were used to evaluate designs *in silico*. Each target design consists of the six conventional LAMP primers: F3, B3, FIP, BIP; LF, and LB (30). The IAC was designed using a composite primer technique (31) for LAMP. IAC template sequence was derived from target region "NC1" by substituting the target loop primer binding sites with engineered sequences. One of the engineered IAC loop sites was used as an IAC loop primer while the other was omitted, so that the IAC assay uses a single loop primer (LFc mut in Fig. 1B). For each primer set a loop primer was modified by the addition of an engineered probe adapter sequence at its 5′ end, with all targets sharing a common adapter and the control assay using a second unique adapter sequence. Primer oligonucleotides were sourced from Integrated DNA Technologies (Coralville, IA, USA).

**Universal displacement probe design.** All oligonucleotides and synthetic targets were purchased from integrated DNA technologies (Coralville, IA, USA). Two engineered universal displacement probes (UDP) corresponding to the target adapter or IAC adapter sequence were designed. Each UDP consists of an oligonucleotide duplex with a 3′ overhang and a fluorophore quencher pair (32). The adapter sequence is located at the 3′ overhang position, with a fluorophore spacer sequence at the 5′ end and a 5′ terminal fluorophore (6-FAM or TEX615). The quencher (Iowa Black FQ or Black hole Quencher-2) sequence is a complementary fluorophore spacer sequence and is labeled with a 3′ dark quencher so that it quenches the fluorophore when annealed. Probe adapters and universal displacement probe sequences were generated from randomized sequence and manually modified in Geneious, using OligoAnalyzer and NUPACK as secondary analysis tools, to minimize dimer and hairpin structures within and between the probes and adapted loop primers. All designs were tested individually and multiplexed against a synthetic dsDNA gBlocks N gene fragment target and ssDNA IAC Ultramer to inform iterative design changes to individual assays. Final design iterations are reported.

**Patient samples.** A panel of 102 human respiratory specimens was used to evaluate our mLAMP assay performance. These specimens collected from nasal or nasopharyngeal swabs were suspended in 3 mL viral transport medium (Becton, Dickinson 220220), aliquoted, and stored at −80°C until testing as described (33). The panel was originally characterized by TaqMan real-time PCR OpenArray plate (ThermoFisher Scientific, Waltham, MA, USA) (34) to contain at least 30 COVID positive samples across a wide range of concentrations and 30 COVID-negative samples as well as other samples identified as positive for other respiratory diseases, including, but not limited to, *Streptococcus pneumoniae*, Influenza, seasonal Coronavirus, Adenovirus, and Enterovirus. Table S1 shows detailed profile in each specimen used in this study. Samples were reassessed in-house for the presence of SARS-CoV-2 RNA, as described below, to account for losses during freeze-thaws, storage, or extraction. In-house results were used as the reference standard. Specimens were collected and tested for SARS-CoV-2 infection as part of the Seattle Flu Study, as approved by the Institutional Review Board at the University of Washington (IRB number: STUDY0006181). Informed consent was obtained for all participant samples, including for use of de-identified, remnant specimens.

**Patient sample preparation.** Specimens were extracted using the QIAamp Viral RNA minikit (Qiagen number 52906) according to the manufacturer's protocol. 100 $\mu$L of sample was mixed with 40 $\mu$L negative VTM (to reach the manufacturer's recommended 140 $\mu$L input), extracted, and eluted in 70 $\mu$L buffer. 5 $\mu$L aliquots were prepared for single use to avoid free thawing and stored at −80°C until use.

**mRT-LAMP protocol.** 20 $\mu$L mRT-LAMP reaction contains 5 mM DTT, 8 mM magnesium sulfate, 20 mM Tris-HCl, 10 mM ammonium sulfate, 10 mM KCl, 0.5% (vol/vol) Triton X-100, 1 $\mu$M each FIP and BIP primers, 500 nM each LF and FB primers, 200 nM each FV and BV primers, 200 nM FAM-tagged UDP probe and TEX 615 UDP probe, 300 nM Quencher 1 and Quencher 2 probes, 10 units of RNasin Plus RNase Inhibitor (Promega, N2611), 6 units of WarmStart RTx (NEB, M0380L), 0.7 $\mu$g TFpol polymerase, and 2 units of thermostable inorganic pyrophosphatase (NEB, M0296L). 5 $\mu$L of extracted RNA was added to 15 $\mu$L mLAMP reaction mixture and incubated at 63.3°C for 1 h on a CFX96 (Bio-Rad Laboratories, Hercules, California). Fluorescence measurements for FAM and TEX 615 signal, indicating SARS-CoV-2 and IAC amplification, respectively, were taken every 25 s (13 s incubation plus a 12 s read). Analysis of the first 40 min (100 cycles) of each run was performed with Bio-Rad CFX Maestro 1.1 software (version 4.1.2433.1219) with FAM channel baseline set as 2 to 35 cycles and a manual threshold of 50 RFU, and Texas Red channel baseline set as 20 to 60 cycles with a manual threshold of 50 RFU.

**RT-PCR protocol.** The RT-PCR protocol was prepared as previously described (32). Each 20 $\mu$L RT-PCR contains 5 mM DTT, 200 $\mu$M ea. dNTP, 1× of either N1, N2, or RP primer/probe mix (IDT, 10006770), 80 mM Tris-sulfate, 20 mM ammonium sulfate, 4 mM magnesium sulfate, 5% (vol/vol) glycerol, 5% (vol/vol) DMSO, 0.06% (vol/vol) IGEPAL CA-630, 8.4% (*wt/vol*) trehalose, 0.05% (vol/vol) Tween 20, 0.5% (vol/vol) Triton X-100, 7.5U reverse transcriptase (NEB M0380L), and 2.5U polymerase (NEB M0481L). 5 $\mu$L of extracted RNA was added to the 15 $\mu$L RT-PCR mixture and subjected to 5 min at 55°C, 1 min of 94°C and 45 cycles of 1 s 94°C and 30 s at 57°C and read using FAM channel on a CFX96. Each clinical sample was run with one technical replicate for each N1, N2, or RP assay, along with standards using synthetic RNA templates prepared in-house and quantified using ddPCR as described (35). Cq and SQ values were exported from Bio-Rad CFX Maestro 1.1 software using the RFU threshold of 50 across all data sets.

**Sequence analysis.** Genomic sequences of SARS-CoV-2 were downloaded from GISAID.ORG (acknowledgments: Tables S2–S9). Criteria for inclusion were: sequences with designation as a Variant of Concern (VOC) or Variant of Interest (VOI) filtered for completeness, high coverage, collection on or before June 14, 2021 and submitted prior to July 1, 2021 and Omicron variant sequences were collected on or before February 1, 2022 and submitted prior to February 15, 2022. The first 1,000 sequence records for each VOC/VOI in the GISAID.ORG database were used for subsequent analysis. This sequence library was screened for perfect identity matches with the primer binding regions of the NC1, NC2, NC3 assays using the packages Biostrings (36) and Seqnir for R (37). The filtration criteria do not omit all sequences with ambiguities (N residues) in LAMP target regions, so mismatches are expected to be slightly overrepresented.

## SUPPLEMENTAL MATERIAL

Supplemental material is available online only.

**SUPPLEMENTAL FILE 1**, PDF file, 2.9 MB.

## ACKNOWLEDGMENTS

We thank other project members in the Lutz lab for their feedback: Robert Atkinson, Michael Roller, Crissa Bennett, Daniel Lyon, and Jack Henry Kotnik. We thank Syamal Raychaudhuri, Frances Chu from Inbios International, and Gwong-Jen J. Chang for stimulating discussion. We thank the Seattle Flu Study and the Seattle Coronavirus Assessment Network (SCAN) teams led by Principal Investigators: Helen Y. Chu, Michael Boeckh, Janet A. Englund, Michael Famulare, Barry R. Lutz, Deborah A. Nickerson, Mark J. Rieder, Lea M. Starita, Matthew Thompson, Jay Shendure, and Trevor Bedford, for providing specimens for testing. We thank GISAID contributors for sequence data (Tables S2-S9).

This work was supported by the Seattle Flu Study (funded by Gates Ventures) and the National Institutes of Health (R01AI145486; 5R61AI140460-03). I.T.H. was supported in part by the National Institute of General Medical Sciences of the National Institutes of Health under Award Number T32GM008268. This work does not reflect the views of the funders. The funders were not involved in the design of the study, and funders do not have any ownership over the management and conduct of the study, the data, or the rights to publish.

E.K. designed this unique version of LAMP assay and in-house polymerase. E.K., I.T.H., and Q.W. prepared the in-house polymerase and synthetic RNA targets. N.P. and E.K. performed mRT-LAMP experiments. N.P. and A.K.O. performed extraction of clinical specimens, RT-qPCR. E.K. performed the analysis of mRT-LAMP and RT-qPCR. P.D.H. and L.M.A. designed the specimens panel used in this study and characterized these specimens using RT-qPCR and OpenArray. B.R.L. oversaw the study. All authors contributed to writing the manuscript.

Patent applications have been filed on several components of this assay. E.C.K., N.P., Q.W., I.T.H., D.L., and B.R.L. are inventors on one or more provisional patent applications. E.C.K., N.P., Q.W., I.T.H., A.K.O., and B.R.L. have equity in a startup company that licenses this technology.

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
