## [Reviewer comments · Microbiology Spectrum]

Microbiology Spectrum

Multiplex Target-Redundant RT-LAMP for Robust Detection of SARS-CoV-2 Using Fluorescent Universal Displacement Probes

Enos Kline, Nuttada Panpradist, Ian Hull, Qin Wang, Amy Oreskovic, Peter Han, Lea Starita, and Barry Lutz

Corresponding Author(s): Barry Lutz, University of Washington

Review Timeline:

Submission Date:	September 20, 2021
Editorial Decision:	November 5, 2021
Revision Received:	April 17, 2022
Editorial Decision:	April 26, 2022
Revision Received:	May 5, 2022
Accepted:	May 6, 2022

Editor: Rosemary She

Reviewer(s): Disclosure of reviewer identity is with reference to reviewer comments included in decision letter(s). The following individuals involved in review of your submission have agreed to reveal their identity: Xianding Deng (Reviewer #1)

Transaction Report:

DOI: <https://doi.org/10.1128/spectrum.01583-21>

November 5, 2021

Prof. Barry R Lutz
University of Washington
Bioengineering
3720 15th Ave NE
Foege Building, Box 355061
Seattle, WA 98195

Re: Spectrum01583-21 (Multiplex Target-Redundant RT-LAMP for Robust Detection of SARS-CoV-2 Using Fluorescent Universal Displacement Probes)

Dear Prof. Barry R Lutz:

Thank you for submitting your manuscript to Microbiology Spectrum. We would be willing to review a resubmission after modifications. When submitting the revised version of your paper, please provide (1) point-by-point responses to the issues raised by the reviewers as file type "Response to Reviewers," not in your cover letter, and (2) a PDF file that indicates the changes from the original submission (by highlighting or underlining the changes) as file type "Marked Up Manuscript - For Review Only". Please use this link to submit your revised manuscript - we strongly recommend that you submit your paper within the next 60 days or reach out to me. Detailed information on submitting your revised paper are below.

Link Not Available

Sincerely,

Rosemary She

Journals Department
Reviewer comments:

Reviewer #1 (Comments for the Author):

Although the approach of LAMP multiplexing using UDP is interesting (novel), this paper has many issues and some flaws related to the assay.

First, separate RNA extraction is required before RT-LAMP (may take 1 hour manually or using equipment for many samples), so it is not "single-pot" strictly speaking; measurement of FAM/TEX615 signal needs special bulky equipment (also takes time to measure). In the end, it may not save much time compared to qRT-PCR. LAMP-based SARS-CoV-2 assay has benefit of portability and point-of-care potential, but this design seems only feasible for use in big labs. How about the cost of this LAMP (seems not low)? Suspect what benefits of this LAMP assay has over conventional PCR.

Second, it looks like this RT-LAMP measures end-point fluorescence or same as qRT-PCR with dynamic curve; if similar to PCR, can Ct value be determined for RT-LAMP assay? RFU cut off is not clear for RT-LAMP, what level considered as negative (below 50)?

Third, limit of detection is not done properly, probit analysis should be applied if this assay wants to be used in clinical labs. In

Fig. 2, more replicates and dilutions are needed (using 3 genes mixed or intact viral RNA) in order to derive LOD at 95%, at least 10 replicates (best 22) with dilutions of RNA, 200, 100, 50, 20, 10, 5, 2.5, 1 and 0. If authors use 96-well plate (high-throughput), this can be done in one plate. Actually authors did not mention if this LAMP can be done in high-throughput plate fashion.

Fourth, LAMP has cross-contamination issue to address, not clear how experiments were done in Fig 3b. Authors should test samples on same plate with some high viral titer and negatives (processed at same time from extraction, ideally blinded), rather than separately tested. More positive samples are needed for robust validation of performance (current 21).

Lastly, IAC (as negative control) is useful, however the system lacks a sample matrix control like RP, so no way to know if NP sample is low quality; this potentially leads to false negatives since RT-LAMP may report as negative due to sub-optimal sample (but LAMP has no quality control for this). Is there any RP-failed samples (by PCR) for testing?

Minor comments:

1. I wonder if some mutations in N gene such as position 28881 may affect this assay performance
2. The sensitivity should be reported as 90% in Abstract, not 100%; specificity of 100% is a bit suspicious, as UDP may have non-specificity. Maybe authors did all negatives together (if in this case), they should validate cross-contamination in plate format, mimicking clinical testing (in blind, high titer could be processed with low titer or negative samples together).
3. Line 53. "...diversity has rendered some NAATs susceptible to false negative results," can authors explain to readers why genetic diversity lead to false negatives, maybe due to mutations in the target region of an assay, is it more susceptible for single gene target than multiple target design; give some examples of commercial assay affected by mutations (citations), e.g. S-drop out by B.1.1.7 using TaqPath.
4. Line 117. What is synthetic dsDNA gBlocks, N gene only, specify.
5. Line 126. Give details of OpenArray, is it Taqman Realtime PCR, which commercial qPCR kit was used or in-house developed. Also add columns of Ct value of N1 and N2 in Table S1.
6. Fig 3A is confusing, above 35 minutes are considered "Not detected"?

Reviewer #2 (Comments for the Author):

In this paper, Kline et al. established a multiplex target-redundant RT-LAMP assay for SARS-CoV-2 detection using universal displacement probes, which enables the detection of various variants. My comments and concerns are:

1. Whether the on-going generation of various variants of SARS-CoV-2 affects diagnostic tests by RT-qPCR and/or RT-LAMP assays should be evaluated (or discussed). Are there any data or literatures to support this view?
2. Multiplex RT-LAMP system based on displacement FIP/BIP probes have been developed (BioTechniques 2012, 53:81-89; Anal. Chem. 2016, 88, 3562–3568). In this study, the authors using universal displacement probe linking/binding to loop primer. In fact, this strategy (based on loop primers) was not novel (had been published previously (Anal. Chem. 2018, 90, 4741–4748; PLoS ONE 2021, 16(3): e0248042: this paper focused on SARS-CoV-2 detection). The authors did not mention the previous papers. On the other hand, the principle is similar between the two strategies (based on FIP/BIP and FL/BL primers). The authors should mention and discuss (compare) the advantages and disadvantages of both methods.
3. Mismatch/variant-tolerant LAMP method had been developed previously for efficient detection of highly variable RNA viruses (e.g. HIV-1) (Front Microbiol 2019 , 10:1056; Analyst 2021, 146: 5347-5356), and showed well performance. The authors should mention and discuss (compare) the advantages and disadvantages of two kinds of strategies for detection of highly variable viruses/targets. This method was also used for detection of SARS-CoV-2 (Virologica Sinica , 2020, 35:344-347).
4. The performance of this multiplex RT-LAMP assay on various mismatches between primers and templates should be evaluated like previous studies (Front Microbiol 2019 , 10:1056; Analyst 2021, 146: 5347-5356).
5. In the clinical evaluation experiments, 30 SARS-CoV-2-positive samples were used. First, the ample size was too small. Second, did these samples contain some SARS-CoV-2 variants? Third, clinical samples used in the evaluation should contain at least two-three most common variant. If there is some difficulty in the collection of clinical samples carrying variants, simulated samples with in vitro synthesized/transcribed variants' RNA should be used.
6. In addition, the gold standard by RT-qPCR assay requires simultaneous detection of two different genes (e.g. N, Orf, and/or E) of SARS-CoV-2. In this assay, only one gene was targeted. The authors should discuss this point.
7. Did the authors perform the extraction-free RT-LAMP assay?

Staff Comments:

Preparing Revision Guidelines

To submit your modified manuscript, log onto the eJP submission site at <https://spectrum.msubmit.net/cgi-bin/main.plex>. Go to Author Tasks and click the appropriate manuscript title to begin the revision process. The information that you entered when you first submitted the paper will be displayed. Please update the information as necessary. Here are a few examples of required

updates that authors must address:

Please return the manuscript within 60 days; if you cannot complete the modification within this time period, please contact me. If you do not wish to modify the manuscript and prefer to submit it to another journal, please notify me of your decision immediately so that the manuscript may be formally withdrawn from consideration by Microbiology Spectrum.

Response to reviewers

Reviewer #1 (Comments for the Author):

(1) First, separate RNA extraction is required before RT-LAMP (may take 1 hour manually or using equipment for many samples), so it is not "single-pot" strictly speaking; measurement of FAM/TEX615 signal needs special bulky equipment (also takes time to measure). In the end, it may not save much time compared to qRT-PCR.

"Single-pot" in this context (line 76) refers to multiplex reactions without physical compartmentalization. Other references to "single-pot" have been removed. This portion of our work does not cover our proposed platform instrumentation, for more on this topic see Panpradist (14). We have emphasized this manuscript's relationship to its companion publication to clarify focused scope of this work.

See:

"This effort serves as the molecular assay basis for our development of a POC diagnostic platform for SARS-CoV-2 (14)." (Lines 90-91)

"Sampling, storage, and portable device solutions with additional clinical and experimental evaluation are already under development (14)." (Lines 390-391)

(2) LAMP-based SARS-CoV-2 assay has benefit of portability and point-of-care potential, but this design seems only feasible for use in big labs.

The methodology, as presented here, would require a well-developed laboratory infrastructure. We believe it is readily adaptable to this environment, as its requirements are not specific to any particular real-time amplification platform. We agree that POC applications are the areas of highest potential impact, and this shared perception is influencing our current and prior work.

See:

"While this proposed chemistry is amenable to adaptation to most contemporary high throughput NAAT platforms as is, it is probably best leveraged in a mobile low resource platform. Sampling, storage, and portable device solutions with additional clinical and experimental evaluation are already under development (14)." (Lines 388-391)

(3) How about the cost of this LAMP (seems not low)?

As presented here, the assay chemistry itself has a cost of about \$5.50/rxn with research scale ordering, however, cost is not a topic of this manuscript. For a thorough breakdown of materials costs, see Panpradist (14).

(4) Second, it looks like this RT-LAMP measures end-point fluorescence or same as qRT-PCR with dynamic curve; if similar to PCR, can Ct value be determined for RT-LAMP assay? RFU cut off is not clear for RT-LAMP, what level considered as negative (below 50)?

The analysis for this assay uses the described real-time parameters for the CFX software. This information has been relocated from supplemental information to the main text, with additional edits. We do not advocate using threshold time of isothermal methods as a means of sample pathogen load estimation.

See:

“Fluorescence measurements for FAM and TEX 615 signal, indicating SARS-CoV-2 and IAC amplification, respectively, were taken every 25 seconds (accounting for 13 second cycle and read times). Analysis of the first 40 minutes (100 cycles) of each run was performed with Bio-Rad CFX Maestro 1.1 software (version 4.1.2433.1219) with FAM channel baseline set as 2-35 cycles and a manual threshold of 50 RFU, and Texas Red channel baseline set as 20-60 cycles with a manual threshold of 50 RFU.” (Lines 154-159)

(5) Third, limit of detection is not done properly, probit analysis should be applied if this assay wants to be used in clinical labs. In Fig. 2, more replicates and dilutions are needed (using 3 genes mixed or intact viral RNA) in order to derive LOD at 95%, at least 10 replicates (best 22) with dilutions of RNA, 200, 100, 50, 20, 10, 5, 2.5, 1 and 0. If authors use 96-well plate (high-throughput), this can be done in one plate.

We have clarified the intent of this manuscript, which does not represent a complete end-to-end diagnostic.

See:

“This effort serves as the molecular assay basis for our development of a POC diagnostic platform for SARS-COV-2 (14)” (Lines 90-91)

“Sampling, storage, and portable device solutions with additional clinical and experimental evaluation are already under development (14).” (Lines 390-391)

(6) Actually authors did not mention if this LAMP can be done in high-throughput plate fashion.

We have added a brief statement to address this comment.

See:

“While this proposed chemistry is amenable to adaptation to most contemporary high throughput NAAT platforms as is, it is probably best leveraged in a mobile low resource platform.” (lines 388-390)

(7) Fourth, LAMP has cross-contamination issue to address, not clear how experiments were done in Fig 3b. Authors should test samples on same plate with some high viral titer and negatives (processed at same time from extraction, ideally blinded), rather than separately tested. More positive samples are needed for robust validation of performance (current 21).

Further clinical testing is performed in our companion publication (14). We have further clarified that this is not our proposed clinical implementation, but rather work towards that goal.

Refer to responses for questions # 2, 5, 6.

(8) Lastly, IAC (as negative control) is useful, however the system lacks a sample matrix control like RP, so no way to know if NP sample is low quality; this potentially leads to false negatives since RT-LAMP may report as negative due to sub-optimal sample (but LAMP has no quality control for this). Is there any RP-failed samples (by PCR) for testing?

This is true, we have included language relating to an endogenous control in our discussion of future work. Note that RnaseP itself is a poor indicator of sample quality and is thus insufficient itself (See “DNA

Cross-Reactivity of the CDC-Specified SARS-CoV-2 Specimen Control Leads to Potential for False Negatives and Underreporting of Viral Infection”, Rosebrock, 2021).

See:

“With the current ssDNA IAC design sampling efficacy and RNA integrity are not assessed and are essential functions for a complete NAAT diagnostic control system. Future iterations will address this by implementing an endogenous human control target and/or encapsidated RNA, such as MS2 coliphage for this purpose.” (Lines 385-388)

Minor comments:

1M. I wonder if some mutations in N gene such as position 28881 may affect this assay performance

We added in additional experimental data relating to the tolerance of the system to known mismatch mutations present in SARS-COV-2 variants. This did not assess 29991 specifically, as that mutation does not interact with the designs presented here.

See:

“Performance against variant sequences with known mismatch mutations

To practically assess the viability of multiplexed target redundancy as a strategy to mitigate diagnostic failure due to primer-template mismatches as a result of viral mutation, the sub-assays and mRT-LAMP were challenged with templates known to have mutations in target regions. Synthetic RNA templates representative of SARS-COV-2 variants Delta (Twist Bioscience, South San Francisco, California, 104539) and Omicron (Twist Bioscience, 105204), as well as the reference genome (Twist Bioscience, 102024) (**RefSeq**) were selected. Delta and Omicron were chosen because of their relative importance to public health (29) and the presence of fixed mismatched mutations in the NC3 and NC1 sub-assays (table 2), respectively. Sequence alignments with Delta sequences identified a high frequency single nucleotide polymorphism (SNP) mutation in that lineage (G29402T) within the NC3 B3 primer binding site. Alignments with Omicron revealed two mismatches in the NC1 assay: a SNP (C28311T) in the F2 primer binding site and a 9-base deletion (GAGAACGCA28362-----) in the NC1 primer binding site. Excepting those conflicts, all other assays were perfect identity matches across their primer binding regions. RT-LAMP sub-assays were evaluated individually against these 200 copies of each template (see Supplemental methods). The NC1 sub-assay failed to detect the Omicron template, while all other sub assays and multiplex assays were successful at detection of all three templates, including Omicron (table 3).” (Lines 283-299)

See:

Added Table 3 (line 301)

See:

“This was further reinforced by directly challenging each of the sub-assays and the multiplex with synthetic RNA templates representative of the reference, Delta, and Omicron lineages. The Delta template was detected by all sub-assays and multiplex assays despite a mismatch in the NC3 B3 primer binding region. This mismatch affected a “bumper” primer less critical to the amplification process, and occurred internally, so tolerance to this low-risk mutation was not unexpected. By contrast, a mutation of the Omicron variant is very high risk; the 9-base deletion to the critical F1 primer binding site would be expected to completely disrupt the loop forming process essential to LAMP. As expected, the NC1 sub-

assay failed to detect the Omicron template when used alone. However, the multiplexed assay was unaffected in its ability to detect Omicron.” (Lines 362-370)

2M. The sensitivity should be reported as 90% in Abstract, not 100%; specificity of 100% is a bit suspicious, as UDP may have non-specificity. Maybe authors did all negatives together (if in this case), they should validate cross-contamination in plate format, mimicking clinical testing (in blind, high titer could be processed with low titer or negative samples together).

A valid point, we have added the sensitivity of 87% to the abstract. Our data on the specificity of this chemistry stands as reported and is not intended to be an evaluation of systemic platform or user specific cross-contamination risk. See prior responses relating to implementation in a clinical setting.

See:

“When challenged with extracted human samples the multiplexed assay showed 87% or better sensitivity, with 100% sensitivity for samples containing greater than 30 copies of viral RNA per reaction, and 100% specificity.” (Lines 29-31)

3M. Line 53. “..diversity has rendered some NAATs susceptible to false negative results,” can authors explain to readers why genetic diversity lead to false negatives, maybe due to mutations in the target region of an assay, is it more susceptible for single gene target than multiple target design; give some examples of commercial assay affected by mutations (citations), e.g. S-drop out by B.1.1.7 using TaqPath.

We have added further language regarding this mechanism in the manuscript, including some specific discussion of a mutation found to affect sub-assay NC1. References (4, 5, 6, 7, others) describe this risk.

See:

“The emergence of this genetic diversity has rendered some NAATs susceptible to false negative results as a consequence of mismatches between their primers and mutations in the targeted nucleic acid, causing these tests to be altered or withdrawn by the U.S. FDA (4).” (Lines 52-55)

“By contrast, a mutation of the Omicron variant is very high risk; the 9-base deletion to the critical F1 primer binding site would be expected to completely disrupt the loop forming process essential to LAMP.” (Lines 366-368)

4M. Line 117. What is synthetic dsDNA gBlocks, N gene only, specify.

Edit made.

See:

“All designs were tested individually and multiplexed in combination against synthetic dsDNA gBlocks™ N gene fragment target and ssDNA IAC Ultramer™ fragments to inform iterative design changes to individual assays.” (Lines 123-125)

5M. Line 126. Give details of OpenArray, is it Taqman Realtime PCR, which commercial qPCR kit was used or in-house developed. Also add columns of Ct value of N1 and N2 in Table S1.

Edits made, including a reference to the original characterization (21).

See:

“These specimens collected from nasal or nasopharyngeal swabs were suspended in 3mL viral transport medium (Becton Dickinson 220220), aliquoted, and stored at -80°C until testing as described (21). The panel was originally characterized by TaqMan real-time PCR OpenArray plate (ThermoFisher Scientific, Waltham, MA, USA) (22) to contain at least....” (lines 128-131)

See:

Supplementary Table S1

6M. Fig 3A is confusing, above 35 minutes are considered "Not detected"?

We have modified Figure 3 to clarify that it is ordinal data (“Positive” and “Negative” categories with copy number data for Positive vs. “Detected” and “Not Detected” categories with Threshold time data for detected IAC and target). The accompanying figure caption has also been revised,

See: Figure 3 and its caption

“Figure 3: mRT-LAMP amplification of extracted nasal specimens. Samples confirmed as Negative or Positive for SARS-CoV-2 by RT-PCR panel (N1, N2, RP) were amplified by duplicate mRT-LAMP reactions. mRT-LAMP signals for SARS-CoV-2 are shown in blue, and IAC signals are shown in orange with detected Threshold time (Tt) or “Not Detected” reported for both in all reactions; replicate pairs for each signal are connected by a line segment. Mean copy number was derived from qPCR results of N1, N2 PCR (see **Supplemental Table S1**).” (Lines 269-274)

Reviewer #2 (Comments for the Author):

1. Whether the on-going generation of various variants of SARS-CoV-2 affects diagnostic tests by RT-qPCR and/or RT-LAMP assays should be evaluated (or discussed). Are there any data or literatures to support this view?

We agree that additional information was needed on this topic. We added in additional experimental data relating to the tolerance of the system to known mismatch mutations present in SARS-COV-2 variants.

See:

“Performance against variant sequences with known mismatch mutations

To practically assess the viability of multiplexed target redundancy as a strategy to mitigate diagnostic failure due to primer-template mismatches as a result of viral mutation, the sub-assays and mRT-LAMP were challenged with templates known to have mutations in target regions. Synthetic RNA templates representative of SARS-COV-2 variants Delta (Twist Bioscience, South San Francisco, California, 104539) and Omicron (Twist Bioscience, 105204), as well as the reference genome (Twist Bioscience, 102024) (RefSeq) were selected. Delta and Omicron were chosen because of their relative importance to public health (29) and the presence of fixed mismatched mutations in the NC3 and NC1 sub-assays (table 2), respectively. Sequence alignments with Delta sequences identified a high frequency single nucleotide

point mutation (SNP) mutation in that lineage (G29402T) within the NC3 B3 primer binding site. Alignments with Omicron revealed two mismatches in the NC1 assay: a SNP (C28311T) in the F2 primer binding site and a 9-base deletion (GAGAACGCA28362-----) in the NC1 primer binding site. Excepting those conflicts, all other assays were perfect identity matches across their primer binding regions. RT-LAMP sub-assays were evaluated individually against these 200 copies of each template (see Supplemental methods). The NC1 sub-assay failed to detect the Omicron template, while all other sub-assays and multiplex assays were successful at detection of all three templates, including Omicron (table 3).” (Lines 283-299)

See:

Added Table 3 (line 301)

See:

“This was further reinforced by directly challenging each of the sub-assays and the multiplex with synthetic RNA templates representative of the reference, Delta, and Omicron lineages. The Delta template was detected by all sub-assays and multiplex assays despite a mismatch in the NC3 B3 primer binding region. This mismatch affected a “bumper” primer less critical to the amplification process, and occurred internally, so tolerance to this low-risk mutation was not unexpected. By contrast, a mutation of the Omicron variant is very high risk; the 9-base deletion to the critical F1 primer binding site would be expected to completely disrupt the loop forming process essential to LAMP. As expected, the NC1 sub-assay failed to detect the Omicron template when used alone. However, the multiplexed assay was unaffected in its ability to detect Omicron.” (Lines 362-370)

2. Multiplex RT-LAMP system based on displacement FIP/BIP probes have been developed (BioTechniques 2012, 53:81-89; Anal. Chem. 2016, 88, 3562–3568). In this study, the authors using universal displacement probe linking/binding to loop primer. In fact, this strategy (based on loop primers) was not novel (had been published previously (Anal. Chem. 2018, 90, 4741–4748; PLoS ONE 2021, 16(3): e0248042: this paper focused on SARS-CoV-2 detection). The authors did not mention the previous papers. On the other hand, the principle is similar between the two strategies (based on FIP/BIP and FL/BL primers). The authors should mention and discuss (compare) the advantages and disadvantages of both methods.

BioTechniques 2012, 53:81-89; was originally cited (ref 10). The QUASAR system (Anal. Chem. 2016, 88, 3562–3568) is an endpoint detection system, and is mechanistically similar/derivative of Kubota (2011) (ref#33). We have included a brief discussion of related real time probe systems and referenced a thorough review.

See:

“A variety of fluorescence probes systems have been previously described (13) and this remains an ongoing area of LAMP innovation.” (Lines 83-84)

“The UDP probe system that enables differentiable detection of Target and IAC amplifications has many commonalities with existing probe systems, but has a unique feature set useful for rapid development, pooled target amplification reporting, and flexible application. Many of the previously described probe

systems, such as DARQ (10) and Molecular Beacons (32) are specific to endogenous target sequence and require dedicated probes. This is also true of assimilating probes (33), the technology that is conceptually most similar to UDPs. UDPs leverage a similar design, exploiting the existing compatible functionality of the loop primers, but use an adapter intermediate so that probe sequences are not directly tied to the target sequence and can be entirely engineered. This provides several advantages: the probes can be designed to be minimally interactive with other elements of the amplification mix, they can be repurposed or adapted to new or revised designs without the need to develop a new probe, and in a multiplex reaction multiple targets can be efficiently associated with a single reporter probe. Mediator displacement probes (34) share these properties, but do not incorporate the probe label into the amplicon which may limit some applications, requires an additional Mediator oligonucleotide, and has a more complex dual labeled stem-loop probe structure. While each of the previously described probe systems has been shown to be effective in various contexts, the suitability of UDPs to a multiplexed target redundant system for a rapidly mutating and variable target with an IAC is apparent.” (Lines 327-342)

3. Mismatch/variant-tolerant LAMP method had been developed previously for efficient detection of highly variable RNA viruses (e.g. HIV-1) (Front Microbiol 2019, 10:1056; Analyst 2021, 146: 5347-5356), and showed well performance. The authors should mention and discuss (compare) the advantages and disadvantages of two kinds of strategies for detection of highly variable viruses/targets. This method was also used for detection of SARS-CoV-2 (Virologica Sinica, 2020, 35:344-347).

We originally cited an earlier paper by the same lead author (Ref#37). We have included some commentary about its compatibility, as well as highlighted a specific example where a mismatch tolerant method would fail.

See:

“While altering or updating a single target assay in a relevant timescale to address Omicron would be logistically challenging, **and mismatch tolerant LAMP methods (37) would still be vulnerable to such a large deletion, a multitarget assay is robust to the threat.** In this scenario, the affected primers could be redesigned to be inclusive to the variant, **potentially including other mismatch tolerance strategies,** and phased into the multiplexed assay without disruption to the diagnostic and ensuring no lapse in coverage over time.” (Lines 372-377)

4. The performance of this multiplex RT-LAMP assay on various mismatches between primers and templates should be evaluated like previous studies (Front Microbiol 2019, 10:1056; Analyst 2021, 146: 5347-5356).

See response to reviewer 2 question 1.

5. In the clinical evaluation experiments, 30 SARS-CoV-2-positive samples were used. First, the sample size was too small. Second, did these samples contain some SARS-CoV-2 variants? Third, clinical samples used in the evaluation should contain at least two-three most common variants. If there is some difficulty in the collection of clinical samples carrying variants, simulated samples with in vitro synthesized/transcribed variants' RNA should be used.

See response to reviewer 2 question 1.

See:

“Sampling, storage, and portable device solutions with additional clinical and experimental evaluation are already under development (14).” (Lines 390-391)

6. In addition, the gold standard by RT-qPCR assay requires simultaneous detection of two different genes (e.g. N, Orf, and/or E) of SARS-CoV-2. In this assay, only one gene was targeted. The authors should discuss this point.

The gold standard testing employed here was based on the CDC 2019-Novel Coronavirus (2019-nCoV) Real-Time RT-PCR Diagnostic Panel consisting of two COV targets, both of which are located in the N gene. In this test, both the N1 and N2 targets require a positive to call a positive, but no formal explanation is provided for this requirement in official guidance. We have refrained from commenting on this requirement but adhered to it in our reference testing. This is no longer considered the standard of practice following updates to the EUA CDC approved diagnostic for SARS-COV-2.

7. Did the authors perform the extraction-free RT-LAMP assay?

No, we did not. We have addressed this comment and this limitation of the study in the text by directing the reader to the relevant study.

See:

“The preliminary testing performed suggests that the system is tolerant to inhibitors that might typically interfere with a direct-to-amplification workflow. However, the clinical specimens evaluated here were processed by RNA extraction. In-amplification sample lysis, and testing with human sample matrix is necessary to further validate this strategy. Assessment of direct detection of contrived samples in nasal matrix and appropriate crude clinical samples is a crucial aspect of ongoing work and was explored in our companion publication (14) focused on a POC detection platform.” (Lines 378-384)

April 26, 2022

Prof. Barry R Lutz
University of Washington
Bioengineering
3720 15th Ave NE
Foege Building, Box 355061
Seattle, WA 98195

Re: Spectrum01583-21R1 (Multiplex Target-Redundant RT-LAMP for Robust Detection of SARS-CoV-2 Using Fluorescent Universal Displacement Probes)

Dear Prof. Barry R Lutz:

Thank you for submitting your manuscript to Microbiology Spectrum. As you will see your paper is very close to acceptance. Please modify the manuscript along the lines I have recommended. As these revisions are quite minor, I expect that you should be able to turn in the revised paper in less than 30 days, if not sooner. If your manuscript was reviewed, you will find the reviewers' comments below.

When submitting the revised version of your paper, please provide (1) point-by-point responses to the issues I raised in your cover letter, and (2) a PDF file that indicates the changes from the original submission (by highlighting or underlining the changes) as file type "Marked Up Manuscript - For Review Only". Please use this link to submit your revised manuscript. Detailed instructions on submitting your revised paper are below.

Link Not Available

Sincerely,

Rosemary She

Reviewer (Editor) comments:

Please modify Abstract to include specific numbers of human samples used in determining the sensitivity and the specificity.

Preparing Revision Guidelines

- point-by-point responses to the issues I raised in your cover letter
- Upload a compare copy of the manuscript (without figures) as a "Marked-Up Manuscript" file.
- Each figure must be uploaded as a separate file, and any multipanel figures must be assembled into one file.
- Manuscript: A .DOC version of the revised manuscript
- Figures: Editable, high-resolution, individual figure files are required at revision, TIFF or EPS files are preferred

Please return the manuscript within 60 days; if you cannot complete the modification within this time period, please contact me. If you do not wish to modify the manuscript and prefer to submit it to another journal, please notify me of your decision immediately so that the manuscript may be formally withdrawn from consideration by Microbiology Spectrum.

Editor comment:

- 1) Please modify Abstract to include specific numbers of human samples used in determining the sensitivity and the specificity.

We have modified the language in the Abstract as requested.

See:

“When challenged with extracted human samples the multiplexed assay showed 87% or better sensitivity (of 30 positive samples), with 100% sensitivity for samples containing greater than 30 copies of viral RNA per reaction (of 21 positive samples), and 100% specificity (of 60 negative samples).” (Lines 29-32)

May 6, 2022

Prof. Barry R Lutz
University of Washington
Bioengineering
3720 15th Ave NE
Foege Building, Box 355061
Seattle, WA 98195

Re: Spectrum01583-21R2 (Multiplex Target-Redundant RT-LAMP for Robust Detection of SARS-CoV-2 Using Fluorescent Universal Displacement Probes)

Dear Prof. Barry R Lutz:

Your manuscript has been accepted, and I am forwarding it to the ASM Journals Department for publication. You will be notified when your proofs are ready to be viewed.

Sincerely,

Rosemary She
Editor, Microbiology Spectrum

Journals Department
Supplementary Information 1: Accept